# Closing the Tobacco Treatment Gap: A Qualitative Study of Tobacco Cessation Service Implementation in Community Pharmacies

**DOI:** 10.3390/pharmacy12020059

**Published:** 2024-03-28

**Authors:** Katy Ellis Hilts, Nervana Elkhadragy, Robin L. Corelli, Micah Hata, Elisa K. Tong, Francis M. Vitale, Karen Suchanek Hudmon

**Affiliations:** 1Richard M. Fairbanks School of Public Health, Indiana University, Indianapolis, IN 46202, USA; kaaellis@iu.edu; 2School of Pharmacy, University of Wyoming, Laramie, WY 82071, USA; nelkhadr@uwyo.edu; 3School of Pharmacy, University of California San Francisco, San Francisco, CA 94143, USA; robin.corelli@ucsf.edu; 4College of Pharmacy, Western University of Health Sciences, Pomona, CA 91766, USA; mhata@westernu.edu; 5Department of Internal Medicine, UC Davis, Sacramento, CA 95817, USA; ektong@ucdavis.edu; 6College of Pharmacy, Purdue University, West Lafayette, IN 47907, USA; vitalefm@msn.com

**Keywords:** tobacco cessation, smoking cessation, implementation science, community pharmacy services, qualitative research

## Abstract

Tobacco use remains a leading preventable cause of morbidity and mortality, with pharmacotherapy and counseling recognized as effective cessation aids. Yet, the potential role of pharmacists and pharmacy technicians in tobacco cessation services is underutilized. This study explores the integration of such services in community pharmacies, identifying facilitators and barriers to their implementation. A qualitative study was conducted across seven community pharmacies in California that were affiliated with the Community Pharmacy Enhanced Services Network. Participants included 22 pharmacists and 26 pharmacy technicians/clerks who completed tobacco cessation training. Data were collected through semi-structured interviews, focusing on experiences with implementing cessation services. The analysis was guided by Rogers’ Diffusion of Innovations Theory. MAXQDA software was used for data management and thematic analysis. Sixteen pharmacy personnel participated in the study, highlighting key themes around the integration of cessation services. Compatibility with existing workflows, the importance of staff buy-in, and the crucial role of pharmacy technicians emerged as significant facilitators. Challenges included the complexity of billing for services, software limitations for documenting tobacco use and cessation interventions, and gaps in training for handling complex patient cases. Despite these barriers, pharmacies successfully initiated cessation services, with variations in service delivery and follow-up practices. Community pharmacies represent viable settings for delivering tobacco cessation services, with pharmacists and technicians playing pivotal roles. However, systemic changes are needed to address challenges related to billing, documentation, and training. Enhancing the integration of cessation services in community pharmacies could significantly impact public health by increasing access to effective cessation support.

## 1. Introduction

Tobacco use is the leading known preventable cause of morbidity and mortality in the United States, resulting in enormous, yet largely avoidable healthcare expenditures [1]. Although two thirds of individuals who smoke would like to quit, and a variety of effective treatment options are available, most attempt to quit on their own, without professional assistance [2], and fewer than 5% of unassisted attempts are successful [3,4]. The most effective approaches combine pharmacotherapy with counseling from a health professional [3], yet few patients report receiving assistance with quitting [5].

To address this important gap in care, the pharmacy profession has systematically attempted to equip pharmacists with the necessary knowledge and skills to assist patients with quitting. Efforts over the past 20 years include (a) broadscale implementation of training programs for licensed pharmacists [6,7,8] and pharmacy technicians [9], (b) development and dissemination of a shared tobacco cessation curriculum for schools of pharmacy [10,11], and (c) advancement of legislation permitting pharmacists to prescribe FDA-approved medications for cessation [12].

Published literature and the accessibility of community pharmacies support a role for pharmacists and pharmacy technicians in addressing tobacco use and dependence. Given that 89% of Americans live within 5 miles of a community pharmacy [13], it can be a convenient location for receiving healthcare services. Additionally, pharmacists have been shown to be effective in helping patients quit [14,15,16,17], and they are able to reach uninsured and under-resourced patients, as well as patients living in rural areas who might experience barriers to accessing primary care [18,19].

Although more than two decades have elapsed since pharmacy-targeted efforts were launched, the processes for implementing tobacco cessation services into pharmacy practice have yet to be examined. With the goal of obtaining a robust understanding of associated facilitators and barriers, a qualitative study was conducted with pharmacists and pharmacy technicians who were selected to receive training and provide tobacco cessation services for a multi-site, pilot demonstration project in their community pharmacies.

## 2. Materials and Methods

### 2.1. Study Sites and Eligibility Criteria

Study sites included pharmacies in California affiliated with the Community Pharmacy Enhanced Services Network (CPESN)—a clinically integrated, organization of community pharmacies providing medication optimization and enhanced patient care services across 49 local networks in 44 US states. After consultation with the lead CPESN administrator in California, ten independently-owned pharmacies were identified as potential study sites.

Pharmacy owners at these locations completed a web-based screening survey to assess minimal qualifications for participation in the pilot, which included the ability to (a) require that all staff (pharmacists and technicians/clerks) participate in a tobacco cessation training program, (b) promote tobacco cessation services in the pharmacy through signage and materials, (c) ask patients filling prescriptions if they use tobacco and document tobacco use status in the electronic pharmacy management system, (d) among patients using tobacco, assess and document interest in quitting, and (e) bill for reimbursement for clinical services. Other requirements included staff completion of web-based post-training surveys, staff participation in qualitative research interviews, and pharmacy owner (or designee) participation in weekly web-based meetings with study sites to share best practices. Seven pharmacies met the minimal qualifications and agreed to participate in the pilot. These pharmacies were provided USD 1500 by CPESN through grant funds for participation in the research component of the pilot study.

### 2.2. Training of Pharmacy Personnel

Pharmacy personnel at each store completed a web-based tobacco cessation training program appropriate for their professional role. Pharmacists (n = 22) participated in a two-hour program derived from the Rx for Change curriculum [11] that addressed the epidemiology of tobacco use, pharmacology of nicotine and principles of addiction, drug interactions with smoking, medications for cessation, assisting patients with quitting (brief counseling promoting referral to tobacco quitlines), and a case-based application of these principles. Technicians (n = 20) and clerks (n = 6) viewed a 30-min program that focused on asking about tobacco use, advising patients to quit, and based on patient preference, referring patients who were ready to quit or would like more information to either the pharmacist and/or to the tobacco quitline [9]. The technician/clerk training also included a brief overview of the medications for cessation. Both training programs provided continuing education credit approved by the US Accreditation Council for Pharmacy Education.

### 2.3. Community of Practice Meetings for Pharmacy Personnel

Two pharmacy faculty members (M.H., R.L.C.) facilitated weekly 30-min web-based group discussions with pharmacy personnel from the seven study sites to discuss facilitators and barriers associated with implementation of tobacco cessation services. Initially, the meetings were primarily attended by the lead pharmacist (i.e., pharmacy owner or manager), with only a few technicians.

Over time, the essential role of the technicians and clerks became increasingly apparent, and each lead pharmacist was asked to designate a technician/clerk “champion” for their pharmacy. This individual was asked to attend the weekly meeting and provide a brief report for their respective pharmacy. Information shared during the meetings included updates on best practices for resolving challenges with implementation, the number of patients with smoking status documented in the pharmacy management system, the number of referrals to the tobacco quitline, success stories with patients, and experiences with billing for services.

### 2.4. Qualitative Interview Approach and Theoretical Framework

Semi-structured interviews were conducted with each participant. The interview guide included main questions, follow-up questions, and probes. Rogers’ Diffusion of Innovations Theory was applied as a guiding framework [20], encompassing five main factors that influence the successful adoption of a new protocol: (a) relative advantage (i.e., the degree to which the new service is perceived by pharmacy staff as better than previous practices); (b) compatibility (i.e., the new service is perceived as being consistent with the values, past experiences, and needs of the community pharmacy setting); (c) complexity (i.e., the degree to which the new service is perceived as difficult to implement and use); (d) trialability (i.e., the degree to which experimentation is possible with the new service); and (e) observability (i.e., the ability to see the impact as a result of implementing the new service).

### 2.5. Data Collection and Analyses

Interviews, designed to be 30 to 40 min in duration, were conducted via Zoom platform with the audio recordings subsequently de-identified and transcribed by a professional service. Participants received a USD 30 Amazon.com gift card.

Two investigators, K.E.H. and N.E., conducted interviews and performed the qualitative thematic analysis, independently engaging in analysis by reading, re-reading, and listening to the interview recordings [21,22]. Through line-by-line analysis, they identified and labeled significant statements to code the data. Following this, they convened to reconcile their coding approaches, achieve consensus, and develop a codebook to systematically document the identified codes. No new codes emerged after analyzing eight transcripts, signaling the achievement of data saturation. Each identified code was then mapped with one of the dimensions of Rogers’ Diffusion of Innovations Theory [20]. The preliminary findings were shared with the research team for further input. MAXQDA Analytics Pro 2020 software [23] was used for data management and analysis.

## 3. Results

### 3.1. Interview Participants

A subset of pharmacy personnel (n = 16) representing 7 pharmacies were recruited for participation in the qualitative study. This included nine pharmacists (seven owners, two managers) and seven non-pharmacist “champions” (six technicians and one clerk). Fifteen interviews were conducted with these sixteen individuals (one joint interview). Most participants were female (n = 10) and identified as white (n = 7) or Asian (n = 6). Two participants identified as some other race, and one was American Indian or Alaskan Native. The average age was 36.9 years (SD, 9.3; range, 27–56).

### 3.2. Findings

Themes and select representative quotes are presented for Rogers’ Diffusion of Innovations [20] constructs in Table 1 and Table 2. Table 1 includes *compatibility* themes, primarily addressing the “fit” of tobacco cessation services within the pharmacy culture and workflows. Table 2 includes themes related to the four remaining constructs: *relative advantage, trialability, complexity, and observability*. A description of themes, by construct, is presented below.

#### 3.2.1. Compatibility Construct—Pharmacy Culture and Pharmacy Workflow

##### Pharmacy Culture

*Patient population served.* This theme reflected participants’ recognition of the needs of their specific patient populations. Several interviewees reported feeling well-connected to their patients, and this created a level of trust that they felt facilitated the success of the tobacco treatment programs. Others acknowledged specific factors about their patient populations that further influenced the intervention, such as the number of tobacco users. For example, one pharmacy noted that they had a very low smoking prevalence among their patient population, which likely led to fewer opportunities to engage with individuals on cessation.

*Staff buy-in and involvement.* Respondents emphasized that having staff buy-in at all levels was important for ensuring the success and sustainability of the new program. For many of the technicians, being able to take an expanded role in assisting patients was a key motivator in their work. Several respondents indicated that engaging staff and generating excitement about the program were crucial to its success. Some of the cited factors that supported obtaining buy-in included a pharmacy-level commitment to a clinical care model and being able to document success and provide feedback.

It was noted that technicians and clerks were often the natural first contacts with pharmacy patients and as such were typically responsible for initiating the screening process. Respondents described how this reduced the time burden for pharmacists and allowed them to focus on providing counseling and support to assist patients with quitting. Further, it was noted that having a technician-driven program highlighted their potential to support clinical services delivered in the pharmacy.

##### Pharmacy Workflow

*When to ask about tobacco use.* Interviewees noted several points in the workflow that worked best for screening for tobacco use in their pharmacies. Most pharmacies initially began by asking about tobacco use when updating records of existing patients. While this was successful, it also became clear to several pharmacies that they were quickly reaching most of their regular patients. Vaccine intake was another key point in the workflow that was identified as an opportune time to screen and engage patients related to tobacco use. Most pharmacies found that incorporating a question regarding patient tobacco use in the vaccine intake form was an effective and low-burden screening approach. Other key points in the workflow that were reported by pharmacies included prescription pickup, new patient intake, and when discussing other health issues. At least one respondent indicated that their pharmacy had formally incorporated the screening as part of monthly medication synchronization calls with patients, while others suggested that seeing specific types of prescription or over-the-counter medications, such as cough medicine, provided an opportunity to initiate a conversation.

*Initiating tobacco treatment services.* When describing the process for offering new tobacco treatment services, including counseling, and providing medications, interviewees described that this process almost immediately followed a patient who had reported current tobacco use. As described above, this initial screening was commonly completed by a technician or clerk. Several technician interviewees described discussing the importance of quitting with the patient, letting them know about the new services, and offering to connect them with a pharmacist. This step varied by pharmacy, with some reporting that being able to immediately start services for patients interested in quitting on the spot was the most effective. However, others felt that scheduling a follow-up appointment with patients was less likely to interrupt the workflow, ensuring that the pharmacist had ample time to spend with patients.

Several interviewees acknowledged that not everyone who screened positive for tobacco use was interested or ready to engage in cessation services. Many stated that they developed a standard approach for documenting this information and then informed these patients of available services, for when they are ready to quit. Some also proactively tried to re-engage folks at other encounters, noting that they had some patients who did return later.

*Medication and behavioral counseling.* Pharmacists reported differential levels of confidence in discussing cessation options with patients. Some pharmacists indicated high levels of confidence in counseling for nicotine replacement therapy (NRT) due to the available resources, guidelines, and prior training, while others described lesser confidence, especially when encountering challenging patient cases or when patients were candidates for medications that required a prescription (i.e., bupropion and varenicline). In these instances, respondents indicated they referred patients back to their primary care provider. Time constraints and a perceived lack of training in behavioral counseling were also noted as barriers to feeling fully confident in assisting patients with quitting. While most reported providing some level of behavioral counseling as part of their interactions with patients, they typically referred them to the state’s tobacco quitline for this as well. In many of these discussions, interviewees indicated their goal was to “meet the patient where they were” and ensure they obtained the assistance that they needed to quit successfully. When considering the time needed to provide the new tobacco treatment services in the community pharmacy setting, most respondents indicated that the initial encounter was between 15 and 30 min in duration. This varied depending on the pharmacist involved and the needs of the patient. The follow-up conversation was typically 5 to 10 min in duration.

*Follow-up care.* Respondents described different ways that they structured follow-up contacts with patients. Typically, these were conducted within two to four weeks of the initial prescription, with some just being calendar-type reminders and others connecting it to refills for the prescription. The follow-up was typically shorter than the initial visit and was used to assess the status of the quit attempt, offer additional support, adjust medications, or troubleshoot with individuals who had not begun using the products yet.

*Service documentation.* There was variability in what participants reported worked best for their pharmacy to document tobacco cessation interactions with patients within workflows. A few described that they could document tobacco use status and some or all of the cessation services directly into their pharmacy software system. Others indicated they had created their own tracking system using tools, such as Excel, electronic calendars, or web-based (e-care) plans to document tobacco treatment services. Interviewees noted the importance of having this documentation not only to ensure they were providing optimal care to patients but also to demonstrate their role in addressing tobacco use, which is essential to obtaining reimbursement for the delivery of a clinical service.

*Service promotion.* When discussing how pharmacy personnel had promoted their new tobacco cessation service, most respondents indicated that they had not actively been advertising the services. However, several had utilized existing materials, e.g., brochures from groups such as the U.S. Centers for Disease Control and Prevention [24] to highlight the importance of quitting and how their pharmacy could help. Others promoted it to providers in their area or network, posted materials indicating they were a quit-smoking pharmacy, or created social media posts to announce the services.

#### 3.2.2. Relative Advantage Construct

*Benefits of pharmacy-based service.* Participants highlighted myriad insights into pharmacists’ roles, particularly in facilitating smoking cessation efforts among patients. They illustrated how the pharmacist acts as a collaborative, accessible, and informative healthcare provider. Emphasis was placed on the practicality and convenience of leveraging pharmacies as accessible healthcare touchpoints, where they serve as a more immediate and more approachable healthcare professional. Participants expressed the tangible health benefits of pharmacists in impacting healthcare expenditures, e.g., by avoiding costly health complications and reducing medication needs.

*Service initiation/expansion.* Pharmacists described the implementation of a tobacco treatment program in their stores, navigating through the initial challenges of ensuring consistent assessment of tobacco use status. Challenges included the change in workflow and sustaining motivation and engagement among the staff. Interviewees indicated they felt that offering the tobacco treatment service provided a pathway for deeper engagement with patients. Through these new services, pharmacists described a shift from a reactive response to health issues to proactively initiating conversations with patients and guiding them through the available treatment options.

#### 3.2.3. Complexity Construct

*Competing priorities.* This theme captured several challenges experienced by pharmacists and technicians in implementing tobacco treatment programs, such as lack of time, understaffing, busy seasons, and multitasking. Participants expressed their desire to provide in-depth and comprehensive tobacco treatment services, however, the workload could be a hindering factor.

*Patient resistance.* Other challenges uncovered during interviews were concerns regarding patients feeling offended or hesitant when queried about tobacco use, with respondents indicating that some might perceive these questions as an intrusion of privacy. For example, some interviewees described this hesitancy when a patient denied using tobacco but emitted an odor of cigarettes. In addition, some described patients’ commitment fluctuating throughout the process. While they initially showed interest in quitting tobacco, their motivation waned over time.

*Tobacco use documentation field.* Several interviewees detailed their experiences and challenges regarding the documentation of patients’ tobacco use status in their pharmacy software systems. While some systems have integrated fields or categories to document tobacco use, others lack this feature, necessitating workarounds such as manual notes or the utilization of comment sections to verify if the patient was asked about tobacco use. This made it challenging to maintain consistent documentation, avoid repeat questioning of patients, and ensure seamless access to these data for all staff members. Some participants suggested that software improvements, like pop-up reminders or automated questions, would be useful.

*Pharmacy reimbursement.* One of the most frequently expressed concerns among participants was the lack of reimbursement for providing tobacco treatment services in the pharmacy. They described how time-intensive counseling patients to quit is, yet there is no sustainable payment and reimbursement model to justify the time and expertise required.

*Cost to patients.* Participants emphasized that insurance coverage, particularly state-funded programs, such as Medicaid, plays a pivotal role in facilitating patient acceptance and usage of NRTs, with patients being notably more receptive when their co-payment is zero. Conversely, those who need to pay for their NRT out-of-pocket tend to hesitate in their quitting therapy or use the medications sub-optimally (e.g., stretching patches over longer periods, choosing single-agent NRT over a recommended combination). Several interviewees indicated that they felt that the short-term expense of cessation therapy, despite having long-term health benefits and saving money in the long run, was a deterrent to patients.

#### 3.2.4. Trialability Construct

*How to ask about tobacco use.* Determining how to ask patients about their tobacco use status was often described as a dynamic process. Many participants described coming up with an initial idea of how to ask, and then modifying the question over time as they tried different wording. For example, instead of asking a narrow, “Are you a smoker?” question, they experimented with broader questions about any tobacco and/or nicotine product use. Modifications were sometimes because of direct feedback from patients. For example, one participant said, “We started out asking ‘Are you a smoker?’ and found that people would say, oh no I don’t smoke but I vape, or I chew… so we switched it”.

Participants also explored strategies for posing these questions in ways that would be less likely to be perceived as judgmental or as an invasion of privacy. Many interviewees acknowledged that asking about tobacco could be met with resistance or defensiveness from patients; thus, it was important to support staff in navigating how to integrate these questions naturally into existing workflows and protocols, as described in the subtheme “*When to ask about tobacco use*,” above.

*Staff training/practice.* Some interviewees suggested that the integration of cessation measures into the pharmacy workflow initially faced challenges such as inconsistent application and staff forgetfulness of new procedures. Respondents indicated that while there was some initial hesitation, with ongoing training and real-life practice, staff grew more comfortable and efficient in providing these new services. Emphasis was placed on the importance of sufficient training, tactful communication, clear protocols, and perseverance of team members, especially the pharmacy “champions.”

*Speed of implementation.* This theme reflects a general rapidity and immediacy in applying the learned protocols following training. Respondents indicated a varied but generally swift implementation, ranging from immediately after training to within a couple of weeks or a month afterward. This rapid implementation underscores pharmacists’ and staff’s willingness and perceptions of the fit or ease of adapting the new tobacco treatment services into existing workflows. How rapidly the services were implemented was impacted by varying levels of comfort, readiness, and procedural differences among the participating pharmacies.

*System workaround.* Interviewees described alternative strategies to overcome the lack of designated software fields for tracking tobacco use and patient engagement in services. For example, to avoid repetitive questions and to efficiently keep track of those already queried about tobacco use, one interviewee described a system that involved adding an asterisk to patient names to indicate that they had been asked. Another pharmacy used a Google sheet to track appointments with the pharmacist for tobacco cessation counseling.

#### 3.2.5. Observability Construct

*Patients assisted.* Interviewees described the number of patients for whom they had provided cessation services. Many described successful interactions with individuals whom they had identified as tobacco users. A couple of examples included assisting a student to quit vaping after being expelled from school and helping couples jointly navigate smoking cessation. As was described previously, while some patients were not initially ready, providing resources and “check-ins” sometimes led to patients returning for future assistance. Conversely, some also noted that not all of those approached have fully engaged or returned for further help, underlining a gap between initial contact and sustained participation in the tobacco treatment program.

*Data monitoring/feedback.* Interviewees highlighted the importance of utilizing strategies to keep track of patient progress and ensure consistent engagement. Strategies described included deploying reminder systems, taking comprehensive notes on patient statuses and outcomes, and employing bi-weekly reviews of performance metrics with staff. Periodic calls and meetings with the research team and other pharmacies provided an opportunity for learning support, sharing insights, and sustaining motivation among pharmacy team members. Despite the reported self-motivation among staff, many felt that maintaining the momentum of the program required a blend of structured data monitoring, feedback mechanisms, and celebrating wins among team members.

*Patient response/feedback.* Interviewees indicated that, overall, patients responded positively to the tobacco treatment services. Participants described how, for some patients, the pharmacy environment could be more appealing as compared to physicians’ offices. Several described instances of patients expressing gratitude and excitement for the new services, especially regarding access to cessation aids and support to ensure the medications were covered through insurance. For example, one participant described how they helped a patient troubleshoot how to use nicotine gum correctly after struggling initially. Interviewees described that even with intermittent setbacks, patients indicated that they valued the service and that the pharmacy’s efforts yielded positive outcomes, including reductions in tobacco use and complete cessation in some cases.

*Scalable/expansion of services.* Despite recognizing the significance and positive impacts of tobacco cessation services, such as improved patient health and diversification of pharmacy services, respondents were concerned about the tangible return on investment and the service scalability. While interviewees expressed a commitment to continuing the tobacco cessation services and, for some, an interest in expanding to include additional clinical service opportunities, this was tempered by concerns about sustainability given unclear financial viability and the need for robust staffing solutions. The concept of expansion is further complicated by factors such as varying levels of patient engagement and the pharmacies’ capacities to offer and manage such programs effectively over time. Despite these concerns, participants exhibited a hopeful perspective, emphasizing the importance and rewards of providing additional clinical services beyond dispensing and articulating a broad vision for the role of pharmacies in healthcare provision and patient management.

Because a central goal of this study was to gain a robust understanding of the “fit” of tobacco cessation services within existing workflows, it was not surprising that many emergent themes were mapped to the *compatibility* construct. Notably, throughout the implementation process, participants identified key moments and locations in the workflow that provided opportunities to initiate the discussion about tobacco—e.g., asking about tobacco use when new patients present at the pharmacy, updating records for existing patients, when discussing other tobacco-related health issues, and when administering vaccines. Identifying these key “opportune moments” is particularly important, as time constraints have been cited as one of the primary barriers to integrating tobacco treatment services in pharmacy settings [25].
pharmacy-12-00059-t002_Table 2Table 2Themes and representative quotes, by Rogers’ Diffusion of Innovation Theory [20] constructs: *relative advantage*, *complexity*, *trialability*, *and observability*.ThemeRepresentative Quotes***Relative advantage***Benefits of pharmacy-based service“Patients can get [help] where they live, because there are pharmacies everywhere. We’re flexible and that’s good for patients and access.”
“When the patient is ready and they want to commit, it’s good to actually get them the product right then and there…versus a couple of weeks later [when] they might not feel as motivated. It helps a lot for us to have the ability to at least get them started.”
Service initiation/expansion“We jumped in and were changing things as we saw fit. It was a work in progress.”
“We were doing smoking cessation even prior…but it wasn’t as active… If somebody was specifically requesting a pharmacist consultation, then we were providing it and we would prescribe certain products that the patient wanted to start. With this pilot program, it became more active, we were more engaged and so the number of patients getting on to the [tobacco cessation] program whether it was through us, or their primary care physician just increased a great deal.”
***Complexity***Competing priorities“We were understaffed, so every time the patient was interested, it was the manager or [another person in charge]… who would schedule that appointment with the patient and do it on his own.”
“Time constraints would be one [barrier/competing priority] and just general knowledge of how to get people to quit.”
Patient resistance“I would say some customers were a little bit hesitant …’cause they feel offended sometimes, I guess.”
Tobacco use documentationfield“We would put an asterisk by the patient’s name if they’d been asked. That way we weren’t asking them every single time because that would be very frustrating for patients you know…”
“We’re very fortunate that our pharmacy software [vendor] updated their system so we have a method to capture [tobacco use] in the software now. We were previously just marking it on their profile and having to put notes and stuff. Now we have fields that capture it, so it makes it much easier for us to see who’s already been asked, what their status is, and what steps we need to take going forward.”
Pharmacy reimbursement“Our biggest challenge remains reimbursement with the health plans not wanting to pay for it. Or they accept the claim, they process it, but then they actually pay us below the cost of getting the product in here, especially for the patches and the gum.”
“The primary barrier is getting reimbursed for the consultation and the service itself…[not just] the medicine…If we don’t get paid for it we can’t put resources toward it.”
“The consults do take time and there’s no easy way for us to bill for the time right now. Again it’s just pharmacists’ free labor as usual…that would be a hindrance to expanding the program.”
Cost to patients“If they’re [on] Medical or one of the state-funded insurances…it’s covered. It’s free, so then they are more receptive to it. They don’t have to buy it or pay out of pocket.”
***Trialability***How to ask about tobacco use“… after asking if they have any allergies to any medications, automatically [we asked] ‘Do you smoke or use any nicotine products?’…then [patients] were less [likely] to question why you’re asking.”
Staff training/practice“The training helped, but then just getting in there and doing it, really was just what I needed.”
“I feel pretty confident in it. I feel like our whole staff is really confident…we’ve been doing it for 6 months now. It’s just part of our day and part of our habit.”
Speed of implementation “Upon completing the training, we were able to develop our policies…and then we immediately started inquiring with folks.”
System workaround “So it’s just a Google form that we created, then we can just go through and ask the questions and at the end, there’s a section to pick an appointment date with a pharmacist.”
“Our software vendor does not currently have a specific field for tobacco use. However, we created a category within our software, to reflect either yes or no.”
***Observability***Patients assisted“One patient [for whom] we provided the [quit-smoking] brochure came back about a month or so later saying, ‘Hey, I spoke to somebody and they provided me with this brochure. I think I’m ready, I want to talk more about it.’”
“[Some] were not our patients…[they] heard about it from elsewhere and came [for assistance with quitting].”
Data monitoring, tracking, and feedback“Roughly every two weeks, we’ll pull the numbers [to determine the] percentage of patients asked…[to] review with the staff so we see what sort of outreach we have…and remind them of the importance of asking…”
Patient response“They’ve [patients] mostly been all positive. We’ve had quite a few patients cut down dramatically, some even completely get off of tobacco, which is really exciting. Even the ones who have had little slip-ups are still really appreciative that we offered the service.”
Scalable/expansion of services “I think we’ll continue. I’m not too sure about expansion…until we start seeing more [return on investment].”
“Our staff is already trained on the process, and it’s becoming part of our normal process…as much as we ask about patient allergies, now we ask, “Do you use nicotine products or smoke?” I think that going forward…it’s not a problem or hurdle for us to keep this practice in place.”



## 4. Discussion

This study applied qualitative methods to assess pharmacy personnel’s perceptions related to the integration of tobacco cessation services, with prescribing, into independently-owned community pharmacies. The findings characterize the facilitators and barriers experienced when integrating these services into routine practice. Understanding these factors is key to identifying effective implementation strategies to support the broader adoption of tobacco cessation services in community pharmacy practice sites [26].

It was evident that there was a learning curve for staff members as they began to ask about tobacco use, highlighting the necessity for infrastructure within pharmacies to support the integration of these inquiries as part of standard practice. For example, leveraging systematic prompts and strategically integrating standard questioning at specific points within the workflow as well as having a field in the software for tobacco use status, appeared to mitigate some barriers to asking directly about tobacco use. This highlights a need for strategies that “promote adaptability,” to allow pharmacy sites to tailor aspects of the intervention to meet their needs while still maintaining fidelity to the intervention [26]. Additionally, ongoing “centralized technical assistance” and opportunities to “capture and share local knowledge” as part of regular meetings with the research team and other pharmacies further augmented their individual and team efforts. Such support is a unique approach of CPESN; thus, it is not necessarily common practice in other community pharmacies.

Another key factor that was identified as a facilitator of implementation was the buy-in and involvement of pharmacy technicians and clerks. Having non-pharmacist support staff who embraced lead roles for implementing the new service enabled the pharmacists to dedicate more of their time toward advising and counseling patients. Several of the “champions” also noted a sense of satisfaction with their expanded role in improving the health of patients. These findings support the expanded role that pharmacy technicians can play in integrating clinical services, including but not limited to tobacco cessation, into routine practice [27,28,29,30]. Further, this highlights the value of “identifying and preparing champions” to support and drive the implementation of these new services [26]. It is conceivable that with additional training, e.g., to become certified tobacco treatment specialists [31,32], technicians could assume an even more integral role.

While overall, pharmacists and technicians in this study reported that these new services fit well within their pharmacy culture and workflows, they also noted barriers to the sustainability and broader adoption of tobacco cessation services in community pharmacies. At the core of these were pharmacy software systems that did not have designated data entry fields for recording current tobacco use status, did not enable documentation of services provided, and did not support streamlined billing for clinical services. It has been noted elsewhere that while pharmacists possess adequate confidence in providing clinical services, they lack knowledge and confidence related to billing and integration of services into workflows [33]. For pharmacists to be able to practice at the top of their license and within their expanded scope, including prescribing for tobacco cessation medications, strategies are needed to “change record systems” to allow for better data monitoring and to “make billing easier” [26].

This study provided a detailed characterization of the personal experiences of pharmacy personnel as they worked together to implement a new clinical service to assist their patients with tobacco cessation. Strengths of the study include the in-depth qualitative approach with pharmacists *and* pharmacy technicians and the application of a theoretical framework. The study is limited in that participants represented only seven independently-owned pharmacies affiliated with a national network in one state (California). The generalizability of study results across other independently-owned pharmacies cannot be assumed, nor should the results be generalized to chain or ambulatory care/clinic pharmacies. Further, given the expanded role of pharmacy internationally in providing preventive services, including tobacco cessation, it is important to note that some of the findings from the current study might not be pertinent in other countries, given differing cultural and regulatory systems [34]. However, there is some evidence that themes presented in this study, such as insufficient time and reimbursement, may serve as barriers to the delivery of these services in other countries as well [35]. Despite these limitations, the study gleaned useful information regarding facilitators and barriers, which will be used to inform future efforts for expanding pharmacy-based tobacco treatment services.

## 5. Conclusions

The results of this study illustrate the tension between the identified value in offering tobacco treatment services in community pharmacies and the practical constraints faced in everyday operations. The data imply a forward-thinking healthcare model, where the pharmacist plays a fundamental, dynamic role in patient care, health management, and ultimately, in cultivating a healthier population. Despite clear advantages, challenges persist, in particular navigating the complexities of billing and reimbursement, as well as managing roles in a busy pharmacy environment. Integrating pharmacy technicians into the clinical service is essential to the overall process and should be explored in greater depth, in terms of advancing their training and their role even further to assist with tobacco treatment.

## Figures and Tables

**Table 1 pharmacy-12-00059-t001:** Themes and representative quotes for Rogers’ Diffusion of Innovation Theory [20] construct, *compatibility*.

Theme	Representative Quotes
** *Pharmacy culture* **	
Patient population served	“I think we can do more than just typing…and giving them prescriptions, because the pharmacists already have a long history in regard to their medications. We’ve already established a good relationship with them. They know that they can trust us and that we’ll communicate with them.”
Staff buy-in and involvement	“The ability of my staff to get engaged and excited about the program I think is huge. I think it would have been much harder if they weren’t so on board with it.” “Pharmacists in general think that we have to do it all, and I mean I was guilty of this too especially in the beginning, and probably why a lot of my clinical services didn’t take off. How am I supposed to catch them at the counter today, engage them in these services, do the intake, and do the prescribing while also filling prescriptions, and taking phone calls? You just cannot, it’s just not even possible. Our staff can be used in so many ways. Our clerk was the first point of contact that went through the patients and found the ones that were interested, and then my technician did the initial intake, so realistically my only portion with that patient is billing and consulting, which is what I already do anyways.” “This is a very technician-driven initiative. It’s helped to engage our technicians in the clinical process to identify and screen and refer to the pharmacist. [It’s] been really nice that they’ve bought into this project as well.” “It’s nice to be a bigger part of people’s health journeys as opposed to just handing over pills.”
** *Pharmacy workflow* **	
When to ask about tobacco use	“[We] basically made it a routine question…especially if they were new patients, after asking them if they have any allergies to any medications, automatically was, do you smoke or use any nicotine products?” “We’ve been trying to focus on [assessing tobacco use status] more significantly for the last several months, and then also adding that question to the flu shot form helps us to capture a broader array of people.” “We try to identify patients when they come in…when they’re getting loaded into the system. If they’re a new patient, we ask them if they smoke or use any nicotine products. We try to make it very clear and make sure we’re capturing information about vaping and nonsmoking forms of tobacco and nicotine use as well… The clerks and the technicians are asking those questions.” “The other way that we’re catching people is when we’re doing monthly med sync phone calls…We inquire with them, “Do you have any new allergies to any medications? Any changes to your med list? Are you seeing a new doctor? Any changes to your insurance? … Do you use any nicotine products?”
Initiating tobacco treatment services	“Whenever we’re having a conversation with the patient, on the phone or in person, even the cashiers and the technicians are asking the question “Do you use any tobacco products?” and from there we start the education process. We ask them “Would you like a pharmacist to speak to you about smoking cessation?” Then, if they’re interested, the cashiers or the technicians will forward the information to a pharmacist and then we will contact the patient…do the interviewing and then prescribing the appropriate products.” “I think [the appointment-based model] kind of helps both sides. The patient knows when they have an appointment…and they’ll be helped at that time and won’t have to wait too long. And the pharmacist can make sure their workflow is managed appropriately.” “We are lucky and are not too crazy busy, so we can do it right on the spot…for [patients interested in tobacco cessation assistance], the walk-in thing is huge.” “Even the patients that aren’t ready [to quit], we remind them that we offer these services just in case they ever are ready. We have had patients who have come back… they’re like, “Oh I’m not ready,” and then months later come back, “Okay, I’m ready.”
Medication and behavioral counseling	“I feel very confident in the nicotine replacement therapy aspect…I have no qualms there. Diving into the [other] medication…I feel a little bit more hesitant…probably just because I don’t have experience doing it.” “We also always referred them to [the tobacco quitline] for the [behavioral] counseling part of it…that’s what worked best for us.” “I would bring [the tobacco quitline] up to everyone. Not everybody wants to get a third party calling them. So it really depends on the patient’s choice.”
Follow-up care	“We do follow-up monitoring in a couple of weeks, to see if they’re doing okay, if there are any concerns…any type of new motivational interviewing required to get them on it. Even after we prescribe the product…for some reason they will hesitate before they start using the products. “We do continuously monitor…creating a [follow-up] routine. I’m sure it helped with certain patients…to have a pharmacist follow up with them and encourage them. They probably felt, like, ‘Oh, the pharmacist cares; maybe I better keep up with this.’”
Service documentation	“If you don’t document it, then it never happened. It has become part of the culture…the constant documentation of activity so that wherever you leave off, someone else will be able to pick up and finish. We strive for that type of documentation…we’re doing what we are supposed to do if we’re asking to get paid for it.”
Service promotion	“We have brochures from the CDC…we put these at the cash register, so patients can see that we offer [tobacco cessation assistance]. We [market the service] on our website…we also do monthly newsletters…and promote on our Facebook page.” “We had a little poster that we put up that said ‘this pharmacy [can] help you quit smoking’…and people were asking about that.” “We don’t really advertise the service that well…I think working on our advertising a bit more would help to increase the reach that we have.”

## Data Availability

The research data are not publicly available.

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
