# Peer review of "Closing the Tobacco Treatment Gap: A Qualitative Study of Tobacco Cessation Service Implementation in Community Pharmacies"

_pharmacy, 2024, doi:10.3390/pharmacy12020059_

Round 1

Reviewer 1 Report

Comments and Suggestions for Authors

It was a pleasure to read this manuscript - it describes a well thought-out study and is a very well written document. It is very US-focused, but the information presented is valuable and can be applied to other health systems, so I see no need to make any adjustments to the manuscript. Congratulations to the team, and especially the writer, for doing an excellent job on the presentation of this project. 

Author Response

Thank you for the positive reviews of our manuscript. No edits were requested, thus there are no responses. 

Reviewer 2 Report

Comments and Suggestions for Authors

This manuscript is well written and organized. I have only one major objection plus a few minor objections, which all are intended to make this manuscript better aligned to the standards and norms of qualitative analysis and reporting.

 1.       This article is a qualitive analysis, but its presentational mode also leans a little on typical reporting of statistical analysis, especially in the results section that largely consists of short subsections addressing a variety of single matters. In my view, these sections are somewhat lacking in the prime characteristic and quality of qualitative interviews: in-depth analysis. I realize that this may be related to the practical orientations underlying the research problem of this study, and I would also like to add that the application of Rogers’ conceptual topology is interesting and serve to lift the argument to a more theoretical level. However, I think that by incorporating the citations presented in table 1 and 2 into the text rather than presenting them as “table findings” (a la statistical reporting), the argument may gain a better “flow”; it would also contribute to make the sections more coherent and in-depth. (Obviously, tables have its place also in qualitative analyses, but then basically as a tool in conceptual considerations and development, not as a major way to present qualitative data).  

 2.       P2 l46: I have no reason to doubt that the “fewer than 5 % who try to quit on their own are successful” statement is correct in the context of the study it stems from, nor do I doubt that a combination of pharmacotherapy with counselling is an effective way to quit smoking. However, as the international tobacco researcher and editor of the leading field journal Tobacco Control for many years Simon Chapman concluded in a review of the larger review on smoking cessation, “up to three-quarters of ex-smokers have quit without assistance (“cold turkey” or cut down then quit), and unaided cessation is by far the most common method used by most successful ex-smokers (REF: Chapman, S. & MacKenzie, R. (2010). The Global Research Neglect of Unassisted Smoking Cessation: Causes and Consequences. PLoS Medicine 7(2): e1000216. doi:10.1371/journal.pmed.1000216). I am aware that this conclusion is not necessarily good news for pharmacists, but the evidence is compelling and should be mentioned also in this study, as it serves to nuance the assumption that successful smoking cessation presupposes use of pharmacotherapy.

 3.       P3l144 and p4l152: It is unnecessary to repeat the use of the word “qualitative” here. The title has already made it clear that this study is qualitative. (The same goes for the table titles, but as already mentioned the content of these tables must be incorporated in the text).

 4.       The information provided in section 3.1 belongs to main section 2, not 3.

 5.       Given that the data stems from California only, I appreciate the authors addressing the limitation of generalizing their results. Still, as Pharmacy is an international journal with readers from around the world, I would urge the authors to reflect some more on the relevance of their study and their findings for readers outside the US. I’m not primarily thinking about generalising the findings here, rather I’m curious as to whether (and how) the research problem and the findings relate to the cultural and regulatory situation in other countries and other parts of the world.   

Author Response

This manuscript is well-written and organized. I have only one major objection plus a few minor objections, which all are intended to make this manuscript better aligned to the standards and norms of qualitative analysis and reporting.

  1. This article is a qualitive analysis, but its presentational mode also leans a little on typical reporting of statistical analysis, especially in the results section that largely consists of short subsections addressing a variety of single matters. In my view, these sections are somewhat lacking in the prime characteristic and quality of qualitative interviews: in-depth analysis. I realize that this may be related to the practical orientations underlying the research problem of this study, and I would also like to add that the application of Rogers’ conceptual topology is interesting and serve to lift the argument to a more theoretical level. However, I think that by incorporating the citations presented in table 1 and 2 into the text rather than presenting them as “table findings” (a la statistical reporting), the argument may gain a better “flow”; it would also contribute to make the sections more coherent and in-depth. (Obviously, tables have its place also in qualitative analyses, but then basically as a tool in conceptual considerations and development, not as a major way to present qualitative data).  

We appreciate that this reviewer values the use of a theoretical framework to guide our interviews; however, we are unclear as to how or why the paper was viewed as lacking in-depth analysis. Our analysis considered all verbiage in the interviews, used standard approaches for qualitative analyses, and led to the mapping of themes and subthemes to the a prior designated theoretical framework. 

Regarding the suggestion to delete the tables and move the quotes into the text – the literature is mixed regarding the “best” way to present qualitative results, and there are publications that argue in favor of the table format for organizational purposes (e.g., https://doi.org/10.1177/1476127020979329). Use of the table does not, in our view, lead a reader to assume any form of statistical reporting – it is a method for organizing the quotes within the context of a multi-construct theoretical framework, for which most constructs/themes had several associated sub-themes.

For our team, the findings and the complexity of the themes identified always dictate our data presentation approach. For this particular paper, we felt that it is easiest for readers to “appreciate” the theoretical framework and its associated themes/subthemes by using a table format. Note: when downloading the manuscript from the MDPI site, the top black bars precluded the white header font within – this has been fixed (Tables 1 and 2).

  1. P2 l46: I have no reason to doubt that the “fewer than 5 % who try to quit on their own are successful” statement is correct in the context of the study it stems from, nor do I doubt that a combination of pharmacotherapy with counselling is an effective way to quit smoking. However, as the international tobacco researcher and editor of the leading field journal Tobacco Control for many years Simon Chapman concluded in a review of the larger review on smoking cessation, “up to three-quarters of ex-smokers have quit without assistance (“cold turkey” or cut down then quit), and unaided cessation is by far the most common method used by most successful ex-smokers (REF: Chapman, S. & MacKenzie, R. (2010). The Global Research Neglect of Unassisted Smoking Cessation: Causes and Consequences. PLoS Medicine 7(2): e1000216. doi:10.1371/journal.pmed.1000216). I am aware that this conclusion is not necessarily good news for pharmacists, but the evidence is compelling and should be mentioned also in this study, as it serves to nuance the assumption that successful smoking cessation presupposes use of pharmacotherapy.

We appreciate this comment and have added one of Chapman’s newer references, along with one other, to indicate the high prevalence of unassisted quit attempts. See: https://pubmed.ncbi.nlm.nih.gov/24399549/ and  https://pubmed.ncbi.nlm.nih.gov/24333037/

  1. P3l144 and p4l152: It is unnecessary to repeat the use of the word “qualitative” here. The title has already made it clear that this study is qualitative. (The same goes for the table titles, but as already mentioned the content of these tables must be incorporated in the text).

This has been amended as requested; we removed “qualitative” from line 143 under 3.1 and also in the titles for tables 1 and 2 (as stated above, we respectfully disagree with the reviewer regarding moving quotes to the text). 

  1. The information provided in section 3.1 belongs to main section 2, not 3.

Based on convention, we have always included sampling results as part of the Results section; as such, we have retained this information in Section 3.1. 

  1. Given that the data stems from California only, I appreciate the authors addressing the limitation of generalizing their results. Still, as Pharmacy is an international journal with readers from around the world, I would urge the authors to reflect some more on the relevance of their study and their findings for readers outside the US. I’m not primarily thinking about generalising the findings here, rather I’m curious as to whether (and how) the research problem and the findings relate to the cultural and regulatory situation in other countries and other parts of the world.   

We agree with this comment and have made relevant modifications to the text (Pg. 13, lines 442-446) and have added additional citations pertaining to pharmacists’ role in tobacco cessation in non-U.S. countries.